# Hysterolaparoscopy: A Gold Standard for Diagnosing and Treating Infertility and Benign Uterine Pathology

**DOI:** 10.3390/jcm10163749

**Published:** 2021-08-23

**Authors:** Valentin Varlas, Yassin Rhazi, Eliza Cloțea, Roxana Georgiana Borș, Radu Mihail Mirică, Nicolae Bacalbașa

**Affiliations:** 1Department of Obstetrics and Gynaecology, Filantropia Clinical Hospital, 011171 Bucharest, Romania; yassinr@windowslive.com (Y.R.); eliza.clotea@gmail.com (E.C.); roxana_georgiana20@yahoo.com (R.G.B.); 2Department of Obstetrics and Gynaecology, “Carol Davila” University of Medicine and Pharmacy, 37 Dionisie Lupu St., 020021 Bucharest, Romania; mirica_rm@yahoo.com (R.M.M.); nicolaebacalbasa@yahoo.ro (N.B.); 3Department of General Surgery, Emergency Clinical Hospital “Saint John”, 014461 Bucharest, Romania; 4Department of Obstetrics and Gynaecology, Cantacuzino Clinical Hospital, 030167 Bucharest, Romania

**Keywords:** hysterolaparoscopy, hysteroscopy, laparoscopy, infertility, benign uterine pathology

## Abstract

Background: Combined hysteroscopy and laparoscopy is a valuable method for diagnosing and treating infertility and benign uterine pathology. Both procedures are minimally invasive, reliable, and safe, with a low complication rate. Aim: In this review, we expose the efficiency and safety of hysterolaparoscopy in the management of infertility and other benign uterine pathologies. Method: We performed a systematic literature review on several databases: PubMed^®^/MEDLINE, PMC, Crossref.org, and Web of Science in the last 10 years. Inclusion criteria: Women of reproductive age with primary or secondary infertility and/or benign uterine pathology. Exclusion criteria: pre-puberty, menopause, couple with male infertility. Conclusion: Hysterolaparoscopy is a useful tool to assess infertility and simultaneously diagnose and treat pelvic and uterine lesions.

## 1. Introduction

The addressability of patients to medical services is largely represented either by psycho-emotional problems related to infertility or by a series of clinical manifestations caused by benign diseases of the genital area [1,2]. Infertility is one of the three main complaints encountered in gynecology, affecting 186 million people worldwide [3], leading to psychological stress, anxiety, and depression [4]. A literature search revealed that the prevalence of primary infertility was higher than secondary infertility (57.5% versus 42.5%) [5]. With the increase of the reproductive age, more patients associate a series of gynecological diseases. For the correct management of each case, it is essential to rule out endocrine disorders and infections and systemic diseases. The most common conditions responsible for woman infertility are endometriosis, polycystic ovarian syndrome, uterine malformations, endometrial polyps, uterine fibroids, premature ovarian insufficiency [6]. Patients are often interested in resolving infertility problems, while others are firstly interested in treating the gynecological conditions and later consider the infertility problem [1,7].

The time between the minimally invasive surgical resolution of the genital pathology and the decision to procreate can also influence the treatment results. Therefore, pre- and postoperative assessment of antimüllerian hormone (AMH) and follicular reserve play a crucial role.

Hysteroscopy is a minimally invasive procedure that permits the visualization of the endocervical canal, uterine cavity, endometrium, and tubal ostia. At the same time, laparoscopy allows the investigation of uterine, tubal, and ovarian capacity [1].

The frequency of laparoscopic findings (pelvic inflammatory disease, endometriosis) is higher than hysteroscopic ones (polyps, uterine septum) in primary and secondary infertility [8]. In women with primary infertility, laparoscopy detected abnormalities in 35% of cases, and hysteroscopy in 17%; while in patients with primary/secondary infertility, hysterolaparoscopy revealed abnormalities in 26% of cases [9].

Besides hormone, ovarian reserve, and ovulation testing, the noninvasive tests recommended for infertility evaluations are ultrasonography, sonohysterography, MRI, HSG, or HyCoSy [10,11]. Hysterolaparoscopy identifies those pathologies that are usually missed by other imaging methods and can correct them. The indication for the surgical treatment must be carefully thought out due to potential complications, such as decreasing ovarian reserve, postoperative adhesions, or a possible postponement in infertility treatment [12].

Many studies suggest that laparoscopy should be considered for women suffering from unidentified infertility etiology because it improves the overall costs of the lengthy treatment plan and the quality of life of the women who want to conceive [13,14].

The aim of this study was to present the value of HL in the diagnosis and treatment of infertility and benign uterine pathology, in accordance with the current evidence.

## 2. Materials and Methods

An extensive electronic search over 10 years was performed to identify all relevant articles on infertility and its surgical treatment. A systematic review of PubMed^®^/MEDLINE, Crossref.org, PMC, and Web of Science Core Collection databases was conducted until 15 May 2021. The search terms regarded the procedure, indication, and outcomes included “infertility”, “benign uterine pathology”, “hysteroscopy”, “laparoscopy”, “hysterolaparoscopy”. In addition, the reference list of the relevant articles was manually searched for other admissible studies. Only articles written in English were included in this search. The first criteria was the relevance of the title, then the information in the abstract was evaluated, and finally, the full-text version of the remaining articles was examined to determine if they were suitable for inclusion in this study.

More than 151 reports were screened for eligibility according to the topic search. As a result, 19 papers were identified, analyzed, and included in the review. The statistical analysis was completed using Microsoft Excel^®^ 2013 (Microsoft^®^ Corporation, Redmond, WA, USA).

Inclusion criteria: Women of reproductive age with primary or secondary infertility and/or benign uterine pathology. Exclusion criteria: pre-puberty, menopause, couple with male infertility.

## 3. Results

### Study Selection

PubMed^®^/MEDLINE data search for RCT/CT or SR/MA in humans in the last 10 years using the selected keywords: “infertility” AND “hysterolaparoscopy”—20 results, “hysteroscopy and laparoscopy”—8 results “laparohysteroscopy”—8 results.

After the initial systematic electronic search, 151 articles were identified. A total of 83 articles were found using Pubmed and PubmedCentral engine search, while 63 were found using Web of Science Core Collection and Crossref.org. Of these, 80 were excluded because of duplicates, unclear and confusing terminology. The remaining 71 studies were assessed at the abstract level, and 52 were further excluded; for the remaining 19 studies, we performed full-text screening for the quality review and were further analyzed. The selection process is presented in the PRISMA flow diagram (Figure 1).

Distribution by type of anomalies across the selected studies is shown regarding uterine, tubal, and ovarian pathology (Figure 2). Hysteroscopic and laparoscopic findings according to eligible studies are described in Table 1 and Table 2. Several clinical studies have been published regarding the use of HL for female infertility (Table 3).

The largest studies, such as those performed by Ugboaja (cross-sectional survey) [29], Mehta (prospective) [20], and Avula (retrospective) [23], reported bilateral tubal occlusion in 46 (20%) patients, 31 (10.33%) patients, and, respectively, 18 (18%) patients, and unilateral tubal occlusion in 84 (36.52%) patients, 30 (10%) patients, and, respectively, 14 (14%) patients.

Endometriosis-related infertility was highly reported in the studies included in our systematic review. A total of 15 out of 19 studies (78.9%) reported cases of endometriosis. Laparoscopic findings of endometriosis or endometrioid cysts were highest in Ravikanth (retrospective) [31], Kavitha (retrospective) [32], and Sapneswar (prospective) [26], where cases were reported in 7 (29%) patients, 32 (25.6%) patients, and, respectively, 9 (22.5%) patients.

A total of 13 out of 19 studies (68.4%) reported cases of uterine polyps. Hysteroscopic findings of uterine polyps were highest in Ugboaja (cross-sectional survey) [29], Neerja (prospective) [16], and Wadadekar (retrospective) [28], where cases were reported in 46 (20.0%) patients, 20 (10.0%) patients, and, respectively, 4 (9.76%) patients.

Submucosal fibroids were also reported in the analyzed studies. A total of 11 out of 19 studies (57.8%) described findings of submucosal fibroids. Hysteroscopic findings were most seen in Ugboaja (cross-sectional survey) [29], Neerja (prospective) [16], and Sapneswar (prospective) [26], where cases were reported in 37 (16.1%) patients, 20 (10.0%) patients, and, respectively, 4 (10.0%) patients.

Uterine synechiae/Asherman syndrome was also reported in the analyzed studies. A total of 15 out of 19 studies (78.9%) described findings of uterine synechiae. Hysteroscopic findings were most seen in Ugboaja (cross-sectional survey) [29], Neerja (prospective) [16], and Ravikanth (retrospective) [31], where cases were reported in 95 (41.3%) patients, 58 (29.0%) patients, and, respectively, 4 (16.7%) patients.

## 4. Assessment of Tubal Patency Using the One-Step Procedure on Hysteroscopy and Laparoscopic Approaches

Infertility in 30% of cases occurs due to tubal pathology [33,34]. Diagnostic tools for tubal factor infertility are hysterosalpingography (HSG), hysteroscopy combined with laparoscopy (HL) and dye test, and hysterosalpingo-contrast-sonography (HyCoSy) [35].

Chronic endometritis represents a common cause associated with recurrent abortion, implantation failure after in vitro fertilization, and endometriosis. Holzer et al. observed the association between chronic endometritis with endometriosis and/or unilateral/bilateral tubal obliteration. Furthermore, hysteroscopy with endometrial biopsy and laparoscopy with chromopertubation (dye test) are currently used to evaluate infertility [36].

In a prospective study of 103 women, Agrawal et al. concluded that hysterolapa-roscopy followed by a dye test could be a reliable alternative to HSG and also permit resolving the gynecologic conditions in the same surgery procedure, especially tubal affections [37]. Kabadi et al., in a retrospective study of 94 patients, found that 53.1% had primary infertility, and 17.1% had secondary infertility. From 28 (29.8%) patients who came for fallopian tube recanalization, the procedure was successful in 75% of cases, and the tubal permeability was reestablished [22].

According to a study of 193 infertile women by Vaid et al., tuboperitoneal pathology is responsible for 40–50% of infertility cases. Therefore, the hysterolaparoscopic approach is more effective than HSG and should be promoted as the only procedure in selected infertile women [38]. The agreement in diagnosing tubal permeability between these two methods was 74%. HSG has a 34.6% sensitivity in diagnosing unilateral tubal block and 80.6% for bilateral block, while specificity was 89.8% for unilateral block and 81.5% for bilateral ones [39].

In a prospective study on 72 patients, Ott et al. showed that the presence of hysteroscopic tubal flow was a good indicator of tubal patency in HL with chromopertubation [40].

Ludwin et al., in a prospective observational study of 132 infertile women investigated for tubal permeability, showed increased accuracy of HyCoSy/HyFoSy techniques compared to laparoscopy with dye test, considered the gold standard. The most accurate noninvasive method was 2D/3D-HDF-HyFoSy (95.8% accuracy), with insignificant differences from reference laparoscopy, while 2D-air/saline-HyCoSy and 2D/3D-HyFoSy had an accuracy of 84.2% and 91.9%, respectively [41].

In a meta-analysis for the assessment of tubal occlusion in women with infertility/subfertility, Alcazar et al. concluded that 2D-HyCoSy has a similar diagnostic performance to 3D/4D-HyCoSy [42]. Another study revealed that the accuracy of 4D-HyCoSy for diagnosis of tubal occlusion was 88.7% compared with laparoscopy and dye test [43]. HyCoSy can be a reliable alternative to laparoscopy with dye test for the evaluation of tubal patency, with a diagnostic accuracy of 91% in endometriosis patients and 92% in non-endometriosis patients [10].

A laparoscopy is an option in patients with bilateral tubal hydrosalpinx undergoing ART procedures. Proximal tubal occlusion is better than salpingectomy for the management of hydrosalpinx, the latter being associated with a significant decrease in ovarian reserve, anti-Müllerian hormone, and an increase in gonadotropin doses [44]. Hysteroscopic tubal electrocoagulation is a proper alternative when the laparoscopy is contraindicated [45].

In a retrospective review published in 2020 on 455 patients, Ekine et al. investigated the pregnancy outcomes after HL managing endometriosis-related infertility. Patients with a proven complete tubal obstruction or those who did not become pregnant 12 months after the surgery were redirected to an in vitro fertilization (IVF) program. The study concluded that HL is an effective and reliable procedure. Furthermore, it is even more effective when combined with assisted reproductive techniques, improving reproductive performance. The overall pregnancy rate was 81.3% (370/455) [30].

## 5. Hysterolaparoscopy (HL) Is a Diagnostic and Therapeutic Tool for Benign Uterine Pathology

Uterine infertility is associated with either change in the shape of the cavity (congenital or acquired), lesions that occupy space (polyps, fibroids), or adenomyosis, which might cause decreased endometrial receptivity [46].

Several studies have shown that only submucosal or intracavitary fibroids can lower implantation and pregnancy rates [46,47]. Therefore, for the treatment of submucosal fibroids, the management is hysteroscopic myomectomy because medical therapy decreases pregnancy rates and is often associated with suppression of ovulation, reduction of the action of hormones at the receptor level, and interference in the implantation process. Surgical hysteroscopy addresses submucosal fibroids to improve conception and pregnancy rates [48]. In addition, laparoscopic or robot-assisted myomectomy is the current therapeutic option for intramural and subserosal fibroids. The combined adoption of these surgical techniques in the management of fibroids will allow patients to benefit from a minimally invasive approach [49].

Endometriosis at various stages has been associated with a higher frequency of endometrial polyps and increased rates of miscarriage. Two studies performed on 1263 patients, and, respectively, 431 patients, observed that the HL procedure increases pregnancy rates in women with endometriosis [50,51]. Hysteroscopy is helpful in the presence of endometrial polyps or uterine malformations (septal uterus) that may be associated with endometriosis [52].

Adenomyosis, a particular form of endometriosis, causes infertility by affecting pregnancy implantation [53]. The laparoscopic diagnosis is made by direct visualization. If, until recently, adenomyosis was diagnosed only on hysterectomy specimens, research has shown that myometrial biopsy by hysteroscopy and laparoscopy is a method by which this condition can be more frequently diagnosed [54].

HL in women with unexplained infertility may identify gynecological conditions that may require ART. For example, on a clinical trial, De Cicco et al. on 170 women with unexplained infertility found a diagnostic rate of pelvic disease in 49.4% of patients. In contrast, in patients who have not been diagnosed with any pathology using combined HL techniques, it has been shown that ART does not improve the pregnancy rate [55].

## 6. How Can HL Manage Uterine Malformations?

The incidence of Müllerian malformations in the general population is about 5.5% depending on each type of malformation [56]. The prevalence of uterine malformations is 4–8% in infertile women, about 13.3–16.7% in patients with recurrent miscarriage, and 24.5% in those with miscarriage and infertility [56,57,58]. The results of two retrospective studies performed on a large number of patients (4005 and 3811 women, respectively) showed a prevalence of Müllerian abnormalities between 4.4 and 7.5%. The distribution of malformations reported varies 54.2–54.9% for septate uterus, 14.2–15.8% for arched uterus, 10.2–10.7% for bicornuate uterus, 5.8–8.5% for unicorn uterus and 3.4–6.5% for hypoplasia/agenesis [59,60].

Siam et al. revealed that, additionally, the rate of gonadal abnormalities was 1.04% (*n* = 40), and Müllerian duct anomalies (MDA) with gonadal abnormalities were 0.57% (*n* = 22). Of the gonadal abnormalities, 70% were diagnosed with hypoplastic ovaries, 25% with gonadal strips, and 5% with an accessory ovary [60].

The diagnosis and therapeutic resolution of female genital tract malformations (FGTM) are customized to each case related to the patient’s symptoms and desire to have a pregnancy. The challenging diagnosis of the malformation requires a double surgical approach for a better resolution of the case [61].

On a group of 117 infertile women or with a history of recurrent abortions, Ludwin et al. showed that HL allowed the identification of 23 arched, 60 septates, 22 bicornuate, and 12 normal uteri [62].

In the diagnosis of uterine malformations, the 3D-SIS imaging was identical to HL [62]. Another study performed on 61 cases verified the concordance between three-dimensional ultrasound (3D US) and HL according to type and classification of uterine abnormality, highlighting in the case of the septate uterus (sensitivity of 100% and a specificity of 92.3%) [63]. The diagnosis of Müllerian anomalies suspected by hysterosalpingography/MRI and confirmed by the HL was replaced by using a 3D US, which, according to Berger et al., has become a gold standard [64].

The correct management of malformations involves adequate surgery to restore physiology and increase the chances of reproduction [58]. Surgical correction of genital malformations depends on identifying the type of abnormality, the association with other malformations, and may involve a multidisciplinary approach.

Endometriosis is associated in about 15% of cases with uterine malformations (20% in bicornuate uterus/didelphys). Follow-up of these cases revealed the disappearance of endometriosis after the correction of the malformation [65].

Rudimentary uterine horns can rarely be accompanied by pregnancy, with an increased rate of uterine rupture in the second trimester [66]. Surgical conduct is given by laparoscopic resection of the dysfunctional rudimentary horn [67,68], without the influence of pregnancy success rate [68].

The use of laparoscopy is important in these cases to exclude other malformations and perform a safety control of the hysteroscopic interventions. Several cases of uterine malformations with a complete uterine septum have been presented in the literature. Surgical resolution consists of hysteroscopic resection and then unification of the two cavities [69]. In an observational study that includes 26 women with double uterine cavities (22 bicornuate and 4 didelphys uteri) with recurrent miscarriages, Alborzi et al. performed laparoscopic metroplasty with diagnostic hysteroscopy, with a second-look by HL 3 months later [70,71]. Bailey et al. support surgical treatment for acquired defects (fibroids, adenomas, adhesions, and polyps), and congenital malformations (didelphys, septate, unicornuate uteri, except bicornuate uteri), improving the pregnancy rate [72].

In addition, a study by Wang et al. on 190 cases showed that HL is a reliable method for diagnosing and correcting the uterine septum, with a 45.2% pregnancy rate [73]. In a study, Munoz et al. revealed that 33.3% of patients with Müllerian malformations (*n* = 59) became pregnant [59].

HL proves to be a precious method in the diagnosis and in the surgical resolution of the detected malformations.

## 7. The Complementary Role of Hysterolaparoscopy Procedures in the Management of Ovarian Abnormalities

The relationship between ovarian cystic formations and infertility is a topic of discussion regarding their influence on fertility and the therapeutic approach. The goal of the treatment is determined by preserving the follicular reserve. The epidemiological analysis reveals the controversial effect of the surgical treatment concerning the expectation regarding the pregnancy rate. Non-aggressive laparoscopic surgical techniques (laser excision, plasma vaporization) and conservative surgery provide a higher functional volume of ovarian parenchyma [74].

The association of polycystic ovary syndrome (PCOS) and endometrial polyps increases the risk of pre-malignant or malignant forms. This scenario becomes a problem in premenopausal women who want a pregnancy. The reality showed that histology in 97.8% of women showed a benign endometrial pathology, and only 2.2% of women, with pre-malignant or malignant transformations, showed polyps. The role of hysteroscopy in these cases is particularly useful, providing both diagnostic data and especially therapeutic solutions [75].

One study has shown a lower postoperative decrease in AMH in laparoscopic cystectomy for benign nonendometriotic ovarian tumors than in endometriomas. At 3 months, the decrease in AMH was similar between the groups [76].

Laparoscopy performed for benign ovarian pathology may be accompanied by hysteroscopy either in the case of evaluation of the uterine cavity before ART or in the case of associated uterine pathology (synechiae, polyp, fibroids).

In a clinical trial on 550 patients with a persistent ovarian cyst ≤ 5 cm who needed ART, Gomez et al. showed that after ovarian cystectomy, patients had a lower number of oocytes retrieved and a similar clinical pregnancy rate or live birth rate [77].

The spontaneous pregnancy rates in infertile women with endometrioma larger than 3–4 cm showed low recurrence rate in excisional surgery than ablation and drainage [78].

## 8. Hysterolaparoscopy—A Therapeutic Challenge in the Management of Endometriosis

More than 10% of reproductive-aged women are affected by endometriosis which can cause, especially, chronic pelvic pain and infertility [79]. In infertile women, the prevalence of endometriosis varies between 20% and 50%, but the accurate percentage is unknown due to diagnostic difficulties [80].

Endometriosis is frequently observed in young women of reproductive age and consists of ectopic endometrial tissues outside the uterus. Under the hormonal influence, this condition produces a chronic inflammatory reaction, with symptomatology represented by dysmenorrhea, lower abdominal pain, and dyspareunia. Usually, the diagnosis can be made after minimally invasive surgery [81,82].

Endometriosis is a therapeutic challenge depending on the stage of the disease. In mild forms, laparoscopic surgery improves fertility by increasing the pregnancy rate. In moderate–severe forms, surgery is improving the symptomatology. The postoperative pregnancy rates are not known due to the lack of clinical trials [82,83,84].

Endometriosis can affect the quality of oocytes and embryos and negatively influence tubal motility [85,86]. In advanced endometriosis, there are changes in the architecture of the pelvic organs secondary to the adhesion process, affecting the release of oocytes and the transfer to the fallopian tube of sperm [84,86].

In a randomized controlled trial that included 200 women, Shaltout et al. showed that Surgicel is a conservative laparoscopic therapy, which decreases the recurrence risk of endometriomas and maintains the ovarian reserve [87]. Another prospective, randomized, blinded, self-controlled pilot study on a total of 16 patients with bilateral ovarian endometriomas ≥3 cm revealed at the 6-month follow-up that ovarian volume (OV) and antral follicle count (AFC) were higher in the laser-treated group versus the stripping technique [88].

In a prospective randomized clinical trial on 122 patients, Sweed et al. showed that deroofing of endometriomas is a better alternative to laparoscopic cystectomy in order to minimize the effect of the ovarian reserve [89].

ENZIAN and rASRM scores are useful for accurate surgical management planning [90]. The endometriosis fertility index (EFI) classification is the only predictable postoperative pregnancy rate system that allows the orientation towards ART of patients with the most unfavorable postoperative prognosis [91,92].

## 9. Discussion

Nowadays, infertility has become a real health problem that affects over 70 million women worldwide, with major social, cultural, religious, and ethnic consequences [93]. Estimates of the incidence of infertility reported to couples of reproductive age worldwide are about 10–25% (between 48 and 180 million) [94]. Over past decades, hysteroscopy and laparoscopy had proven their efficacity and efficiency and are now both used worldwide, offering better outcomes than previous methods.

The one-step use of hysteroscopy and laparoscopy was described to be very safe and effective in the diagnosis of infertility by Dawle et al. [15] in 2014 and Mehta et al. [20] in 2016, while Nandhini et al. [21] suggested later the same year that operational skills could be improved. Several clinical studies have been published regarding the use of HL for female infertility (Table 3).

Minimally invasive surgery for patients with infertility problems improves the diagnosis rate, the therapeutic success rate regarding the detected gynecological pathology, as well as the success rate of IVF techniques and the quality of life. However, some conditions can endanger the patient’s life and are diagnosed following routine investigations for infertility.

Hysteroscopy can see very small changes in the endometrial cavity thanks to its magnification abilities and offer a therapeutic role. Hysteroscopy is not reliable in the absence of pathological elements evidenced by ultrasound (VCI), MRI, HSG, or HyCoSy. Surgical hysteroscopy helps diagnose and correct intrauterine abnormalities (uterine synechiae, septa, fibroids, polyps, endometrial hyperplasia) and adhesions in the tubal ostiums [95]. The role of diagnostic hysteroscopy prior to ART is not yet defined [96], but sometimes, a diagnostic hysteroscopy should be performed before initial IVF, or especially in case of failure of a previous IVF.

Endometrial polyps negatively influence endometrial receptivity, as demonstrated by a meta-analysis that showed a fourfold increase in the expected pregnancy rate after hysteroscopic polypectomy in patients undergoing intrauterine insemination. A systematic review, which included 8 studies with 2267 patients, showed that resection hysteroscopy for endometrial polyps with an average diameter of less than 2 cm was correlated with a higher pregnancy rate after intrauterine insemination [97]. Thus, three meta-analyses showed a live birth rate (LBR) of approximately 1.3–1.48% [98,99,100], while other studies (two large RCT—TROPHY and inSIGHT) did not show any correlation [101,102].

The meta-analysis conducted in 2020 by Metwally et al., which included 4 RCTs with 442 infertile patients undergoing laparoscopic/hysteroscopic myomectomy, showed that the evidence is limited and of little statistical relevance [103]. Starting from the conclusion of this study, it is essential to determine the effectiveness of myomectomy in infertility associated with fibroids, and more research is needed to establish on extensive clinical trials how myomectomy can influence the pregnancy rate. In another meta-analysis, Bosteels in 2018 found no studies on surgical hysteroscopy in infertile women with submucosal fibroids, intrauterine adhesions, or uterine septum before ART, nor whether hysteroscopic myomectomy improves the pregnancy rate [104].

On the other hand, laparoscopy is a safe procedure with a very low complication rate, and permits the investigation of uterine, tubal, and ovarian capacity, and avoids unnecessary laparotomy. Laparoscopic myomectomy is addressed to the subserosal or intramural fibroid, influencing fertility in case of endometrial involvement [48].

In a recent meta-analysis that included 38 studies (3326 women with anovulatory PCOS and clomiphene citrate resistance), Bordewijk et al. revealed that laparoscopic ovarian drilling decreased the live birth rate compared to the ovulation induction pattern alone [105].

In 2014 Neerja et al. [16] showed that the HL techniques are improving the pregnancy rate. This was later proved by Puri et al. [18] in 2015. One year later, Kabadi et al. [22] suggested using HL as a daycare procedure. Niranjan et al. [19] considered HL to be the gold standard in diagnosing and treating female infertility in 2016. Other studies published between 2017 and 2019 had all shown the same result, considering it very safe and effective, decreasing the need for a repeat procedure. Wadadekar et al. [28] and Ugboaja et al. [29] had published in 2020 that it has an excellent diagnostic purpose and unique opportunity to treat female infertility.

Fertility success rates are even higher when laparoscopy is combined with diagnostic or surgical hysteroscopy. The dual approach is important because the suspicion of various tubal, uterine, ovarian abnormalities observed following diagnostic imaging techniques (ultrasound, MRI) must be resolved. The technique of performing hysteroscopy in the same surgery single-step does not increase the morbidity of the case (on the contrary, it does not require new anesthesia) and shows very good compliance of patients by shortening the duration of investigating infertility and increasing patient satisfaction.

Laparoscopy instead visualizes tubal permeability by chromopertubation, performs peritubal and periovarian adhesiolysis, and treats benign ovarian diseases (cysts, fibroids, teratomas). Differences between surgical techniques on peritoneal adhesions using a CO_2_ laser improve subsequent performance. Tubo-uterine abnormalities contribute to approximately 30–35% of female infertility [95], which requires hysteroscopy in the same procedure.

In patients with preserved tubal permeability, expectant behavior is preferred, while laparoscopy is a better choice in patients with impaired tubal permeability or those preparing for assisted reproduction techniques (ART).

The management of endometriosis-related infertility involves information about the different therapeutic alternatives and their sequencing, recurrence rate, benefits, and risks, especially before surgery [106]. Moreover, there is a growing interest in fertility preservation options in the case of patients undergoing surgery for ovarian endometriosis [107].

In minimal and mild forms of endometriosis, the option is first-line surgery (laparoscopy or HL) prior to ART, which may improve the live birth rate. The strategy of resecting as much endometriotic tissue as possible and then using ART in women who failed in 1 year to have a spontaneous pregnancy after surgery led to a pregnancy rate of 78.8% [81,85,108]. One meta-analysis on 3 RCTs with 528 participants of Bafort et al. [109] and one meta-analysis on 2 RCTs with 442 patients of Jacobson et al. [110] have demonstrated an advantage of laparoscopic surgery compared to diagnostic laparoscopy reported to clinical pregnancy rates. In stages I–II of endometriosis, the analysis showed that laparoscopic treatment increases the pregnancy rate. Moreover, laparoscopic surgery improves the viable intrauterine pregnancy rate, but this is evidence of moderate quality [110]. Other studies have shown a similar pattern that laparoscopy increases the chances of a future pregnancy and live birth for minimal and mild endometriosis [61,111].

In moderate–severe forms of endometriosis, the option is first-line ART with no surgery. In patients with deeply infiltrating endometriosis, the spontaneous pregnancy rate is about 10%; the pregnancy rates after surgery vary from 40 to 85% in case of no colorectal endometriosis resection. Furthermore, in the case of laparoscopic colorectal resection, pregnancy rates range from 47 to 59% [112].

In case of ovarian reserve damage (increasing age and decreasing AMH, antral follicle count), the advice is first-line surgery or IVF, but at this moment, there is not a standard consensus. [83]. Many patients want to know the real chance to conceive a pregnancy, and to reach this goal, they want to know the proper algorithm.

Surgery improves postoperative outcome and reduces the risk of recurrence. The increase in the chances of fertility, whether or not followed by ART, is directly proportional to the patient’s age, surgical accuracy, and to an early stage of endometriosis [110,113].

The surgical management of selected cases is dependent on the patient’s age and clinical picture, the desire for conception, and the association of gynecological diseases. We are often in a difficult position to decide because we must answer a crucial question: which is the first therapeutic step in persistent infertility? After exhaustive literature analysis, we suggest that the gold standard in the selected cases of persistent or unexplained infertility is the following sequence: HL prior ART.

## 10. Conclusions

One-step hysterolaparoscopy or various combinations (laparoscopy and/or hysteroscopy, exploratory, and/or operative) are effective methods for identifying and treating anatomical structural abnormalities related to infertility. In addition, an important social problem remains in infertility stigma, which marks and invalidates the psycho-emotional status of women. Thus, the combined procedures of laparoscopy and hysteroscopy represent a gold standard in the treatment of infertility of various causes.

## 11. Practice Key Points

HL is a safe and effective therapeutic tool for benign uterine pathology and correctable uterine malformations.HL is a daycare procedure for the evaluation and treatment of female infertility.HL can diagnose undetectable imaging disorders in asymptomatic infertile patients or patients with mild symptoms.HL is a useful diagnostic method if the imaging techniques (3D-SIS ultrasound, MRI, or HyCoSy) are not accessible [114].HL is superior to HSG in diagnosing the tubal and uterine pathology but with similar accuracy to HyCoSy.HL is the first-line therapeutic option prior to ART in minimal/mild forms of endometriosis.

## Figures and Tables

**Figure 1 jcm-10-03749-f001:**
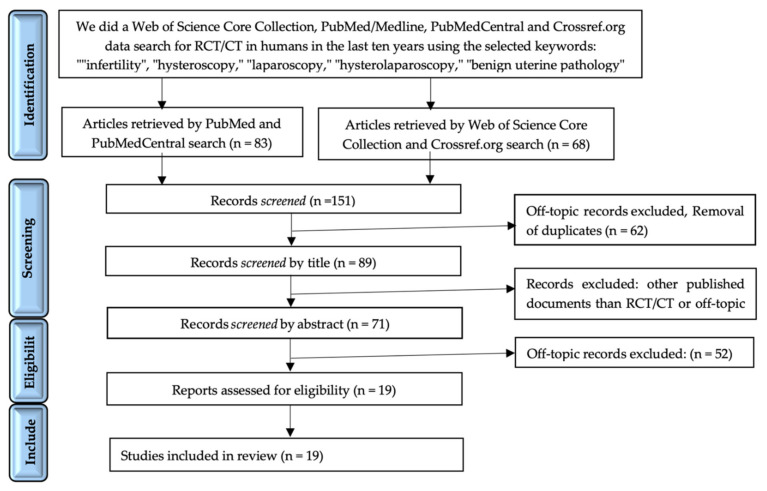
PRISMA diagram—systematic search and study selection process.

**Figure 2 jcm-10-03749-f002:**
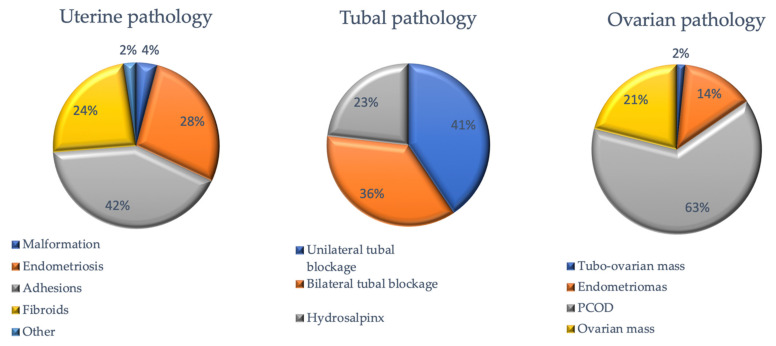
Distribution by type of anomalies on uterine, tubal, and ovarian pathology across the selected studies.

**Table 1 jcm-10-03749-t001:** Hysteroscopic findings.

Study	Uterine Pathology									Tubal Pathology
	Cervical Stenosis	Synechiae	Submucosal Fibroids	Polyps	Asherman Syndrome	Uterine Septum	Malformation	Atrophy	Others	Adhesions in the Tubal Ostium
**Dawle** [15]		7	3	2		1				
**Neerja** [16]		58	20	20			5	25	22	
**Firmal** [17]		1 *	1	1 **						
**Puri** [18]			4			2				
**Niranjan** [19]	6			5	3	2	2		8	2
**Mehta** [20]		1	8	16		29			6	
**Nandhini** [21]	1	1	4	1		1			3	
**Kabadi** [22]		5	6				13			
**Avula** [23]		1	4	6		4		5	14 ***	
**Sharma** [24]		5		10		14				
**Ahmed** [25]		4				4			3 ****	
**Sapneswar** [26]			4			5				
**Shinde** [27]			4	7	1	2	8			
**Wadadekar** [28]		2	1	4		1	1			
**Ugboaja** [29]		95	37	46			34		14 *****	
**Ekine** [30]				28	5	39				1
**Ravikanth** [31]		4				1				
**Kavitha** [32]	2			5			13		3	

*—Intrauterine; **—submucosal; ***—hyperplastic endometrium; ****—ostia not seen; *****—lost iud.

**Table 2 jcm-10-03749-t002:** Laparoscopic findings.

Study	Anomalies												
	Uterine Pathology					Tubal Pathology				Ovarian Pathology			
	Malformation	Endometriosis	Adhesions	Fibroids	Others	Tubal Blockage	Unilateral Tubal Block	Bilateral Tubal Block	Hydro-Salpinx	Tubo-Ovarian Mass	Endometriod Chistae	PCOS	Ovarian Mass
Dawle	2	2	7		2	18	7	11				2	
Neerja													
Firmal		6	7	1				1			1		
Puri		9			3	9			3	1		11	5
Niranjan	2	2	5	4			5	4		2	4	8	11
Mehta	3	41	29 *	15		61	30	31		22			
Nandhini	1	6	5	2 **	1	10	5	5	5	1	4	13	
Kabadi		5	18	6		7	3	4			6	13	
Avula	7	11	12 *	5		32	14	18				48	7
Sharma		9	38	8		12	6	7			10	31	
Ahmed	2		8				3	5			1	4	
Agrawal	67					34				78			
Sapneswar		9				5						9	3
Ravikanth	1	7	8	3			6	6					1
Kavitha G	5	32		17								16	25
Shinde		9	12	4	9 (TB)		6	4					
Wadadekar	2	3	15			18					4	2	
Ugboaja		19	91			130	84	46	96				
Ekine				80			13	20					

TB = tuberculosis; *—adnexal; **—subserous;

**Table 3 jcm-10-03749-t003:** Synopsis of studies.

Nr	Authors	Year	Inclusion Period	Study Design	Age	Nr	Type of Infertility		Hysteroscopy		Laparoscopy		Management		Pregnancy Rate	Conclusion
							P	S	N	A	N	A	D	O		
1	Dawle	2014	2011–2013	PS	18–40	100	100	0	83	13	66	34	D		21%	Very safe
2	Neerja	2014	NR	PS	20–30	200	125	75	50	150	60	140	D	O	45.71%	Improved the rate of pregnancy
3	Firmal	2014	2009–2011	PS	27.6 (mean)	30	24	6	28	2	19	11	D	O	-	Beneficial following failure of empirical treatment in women with unexplained infertility
4	Puri	2015	NR	PS	30 (mean)	50	24	26	44	6	0	50	D	O	28.2%	Higher conception rate Adequate training required
5	Niranjan	2016	2013–2015	PS	20–40	100	87	13	74	26	53	47	D	O	35–45%	The gold standard in diagnosis and treatment
6	Mehta	2016	2013–2015	PS	28.8; 31.1	300	206	94	244	56	199	101	D			Effective diagnostic tool
7	Nandhini	2016	2015–2016	PS	21–40	50	50	0	37	13	43	7	D			Useful—improved with operative procedures
8	Kabadi	2016	2014–2015	RS	18–40	94	50 *	16	43	17	44	47	D	O		Considered a day care procedure for evaluation and treatment of female infertility
9	Avula	2017	2015–2016	RS	20–35	100	72	28	66	34	48	52	D	O		Effective and safe diagnostic tool
10	Sharma	2017	2012–2015	RS	20–40	130	82	48	101	29	48	82	D			Effective, safe, and minimally invasive tool
11	Ahmed	2017	2015–2016	PS	20–40	30	21	9	19	11	17	13	D	O		Very safe and effective
12	Agrawal	2018	2016–2017	PS	19–35 (27.7)	157	93	64			32 HL	125	D	O	57.3%	HL—potential gold standard approach in the evaluation of female infertility
13	Sapneswar	2018	NR	PS	29.5 (mean)	40	24	16			2 HL	38	D	O		Beneficial for the diagnosis and treatment in patients of primary and secondary infertility
14	Ravikanth	2019	NR	RS	20–45	24	21	3	19	5	NR		D	O		Effective, safe, and minimally invasive
15	Kavitha G	2019	2013–2018	RS	19–40	125	104	21	101	24	53	72	D	O		Decrease need for a repeat procedure
16	Shinde	2019	2018–2019	PS	25–38 (33)	100	50	50	80	20	57	43	D	O		Gold standard
17	Wadadekar	2020	2019	RS	20–40	41	32	9	32	9	13	28	D	O	21.05%	Excellent diagnostic purpose and unique opportunity to treat
18	Ugboaja	2020	NR	Cross-sectional survey	35.6 mean	230	106	124	78	152	59	171	D	O		The gold standard for diagnosis and treatment
19	Ekine	2020	2010–2016	RS	25–46; (34.3)	455	319	136	NR	NR	272 HL	168	D	O		Effective and relatively safe procedure; even more effective combined with ART

NR = not reported; PS = prospective study; RS = retrospective study; HL—hysterolaparoscopy; * 26 cases—tubal recanalization; P—primary, S—secondary; N—normal; A—abnormal; D—diagnostic; O—operative.

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
