# Peer review of "Hysterolaparoscopy: A Gold Standard for Diagnosing and Treating Infertility and Benign Uterine Pathology"

_jcm, 2021, doi:10.3390/jcm10163749_

Round 1

Reviewer 1 Report

Manuscript is too long and boring. Introduction in academic way reveal  the problem of infertility and gives well known data upon the causes in percentage and in fact instead of  giving some information about methods of diagnostic and treatment of infertility is a part of results chapter. Authors should decide which figure include in the paper 2 or 3 There is no need to include both. Reviewer suggestion is fig 2.

Section 4  Assessment of tubal patency Were the results of HL compared to HyCoSy? That method of evaluating tubal patency is worldwide used but is not mentioned in the paper.

Section 5 One of the most common benign uterine pathology is adenomiosis. Was this condition evaluated in analysed papers? Line 255 to 259  is not related to this subject and should be excluded.  

Section 7 and 8 and  discussion. Authors should shorten this sections with reasonable conclusions instead of refer well know guidelines and recommendations Discussion  should attempt to critically show the real place and usefulness of laparoscopy and hysteroscopy in treatment of infertility and uterine disorders. It will be appreciated if authors try to make an attempt to find the  conditions where histeroscopy and laparoscopy have reasonable success rate in terms of pregnancy rate when compared to ART and give some recommendations.

Manuscript is too long and boring. Introduction in academic way reveal  the problem of infertility and gives well known data upon the causes in percentage and in fact instead of  giving some information about methods of diagnostic and treatment of infertility is a part of results chapter. Authors should decide which figure include in the paper 2 or 3 There is no need to include both. Reviewer suggestion is fig 2.

Section 4  Assessment of tubal patency Were the results of HL compared to HyCoSy? That method of evaluating tubal patency is worldwide used but is not mentioned in the paper.

Section 5 One of the most common benign uterine pathology is adenomiosis. Was this condition evaluated in analysed papers? Line 255 to 259  is not related to this subject and should be excluded.  

Section 7 and 8 and  discussion. Authors should shorten this sections with reasonable conclusions instead of refer well know guidelines and recommendations Discussion  should attempt to critically show the real place and usefulness of laparoscopy and hysteroscopy in treatment of infertility and uterine disorders. It will be appreciated if authors try to make an attempt to find the  conditions where histeroscopy and laparoscopy have reasonable success rate in terms of pregnancy rate when compared to ART and give some recommendations.

Author Response

Dear Esteemed Reviewer,

Thank you for revising our manuscript.

  1. Manuscript is too long and boring. Introduction in academic way reveal  the problem of infertility and gives well known data upon the causes in percentage and in fact instead of  giving some information about methods of diagnostic and treatment of infertility is a part of results chapter. Authors should decide which figure include in the paper 2 or 3 There is no need to include both. Reviewer suggestion is fig 2.

Answer: Thanks for your remarks; we rewrote the introduction partially, we excluded percentage from the text, and added information about methods of diagnostic and treatment. Also, we included in the paper figure 2 in accordance with your suggestions. (please see the attached manuscript)

  1. Section 4  Assessment of tubal patency Were the results of HL compared to HyCoSy? That method of evaluating tubal patency is worldwide used but is not mentioned in the paper.

Answer: Thank you for your mention; we introduced the main diagnostic tools for tubal patency and compared the accuracy rate between HyCoSy and HL. (please see the attached manuscript). Lines 399-401, 418-463, 467-469.

  1. Section 5 One of the most common benign uterine pathology is adenomiosis. Was this condition evaluated in analysed papers? Line 255 to 259  is not related to this subject and should be excluded.

Answer: Thank you for your indication; we moved adenomyosis in section 5 at benign uterine pathology and analyzed how HL can manage this condition. Also, we excluded lines 255-259. (please see the attached manuscript).

  1. Section 7 and 8 and discussion. Authors should shorten this sections with reasonable conclusions instead of refer well know guidelines and recommendations Discussion  should attempt to critically show the real place and usefulness of laparoscopy and hysteroscopy in treatment of infertility and uterine disorders. It will be appreciated if authors try to make an attempt to find the  conditions where histeroscopy and laparoscopy have reasonable success rate in terms of pregnancy rate when compared to ART and give some recommendations.

Answer: Thank you for your comment; we excluded unnecessary information from sections 7 and 8. In the discussion chapter, we rearranged the information to focus on HL’s roles in diagnosis and treatment of infertility and uterine disorders. Finally, we introduced practice key points. (please see the attached manuscript)

Kindest regards,

Valentin Varlas

From the behalf of authors

Reviewer 2 Report

I congratulate you on the wonderful work done and the excellent method used. I believe there are no notes to be made.

Author Response

Dear Esteemed Reviewer,

Thank you for revising our manuscript and for your consideration of our work.

Kindest regards,

Valentin Varlas

From the behalf of authors